# Peri-Implant Repair Using a Modified Implant Macrogeometry in Diabetic Rats: Biomechanical and Molecular Analyses of Bone-Related Markers

**DOI:** 10.3390/ma15062317

**Published:** 2022-03-21

**Authors:** Hugo Robertson Sant’Anna, Marcio Zaffalon Casati, Mounir Colares Mussi, Fabiano Ribeiro Cirano, Suzana Peres Pimentel, Fernanda Vieira Ribeiro, Mônica Grazieli Corrêa

**Affiliations:** Dental Research Division, School of Dentistry, Paulista University, Av. Dr. Bacelar, 1212, 4° andar, Vila Clementino, São Paulo 04026-002, SP, Brazil; hugorobertson@gmail.com (H.R.S.); marcio.casati@docente.unip.br (M.Z.C.); mounir.mussi@hotmail.com (M.C.M.); fabiano.cirano@docente.unip.br (F.R.C.); suzana.pimentel@docente.unip.br (S.P.P.); fernanda.ribeiro@docente.unip.br (F.V.R.)

**Keywords:** dental implants, gene expression, bone repair

## Abstract

DM has a high prevalence worldwide and exerts a negative influence on bone repair around dental implants. Modifications of the microgeometry of implants have been related to positive results in bone repair. This study assessed, for the first time, the influence of an implant with modified macrodesign based on the presence of a healing chamber in the pattern of peri-implant repair under diabetic conditions. Thirty Wistar rats were assigned to receive one titanium implant in each tibia (Control Implant (conventional macrogeometry) or Test Implant (modified macrogeometry)) according to the following groups: Non-DM + Control Implant; Non-DM + Test Implant; DM + Control Implant; DM + Test Implant. One month from the surgeries, the implants were removed for counter-torque, and the bone tissue surrounding the implants was stored for the mRNA quantification of bone-related markers. Implants located on DM animals presented lower counter-torque values in comparison with Non-DM ones, independently of macrodesign (*p* < 0.05). Besides, higher biomechanical retention levels were observed in implants with modified macrogeometry than in the controls in both Non-DM and DM groups (*p* < 0.05). Moreover, the modified macrogeometry upregulated OPN mRNA in comparison with the control group in Non-DM and DM rats (*p* < 0.05). Peri-implant bone repair may profit from the use of implants with modified macrogeometry in the presence of diabetes mellitus, as they offer higher biomechanical retention and positive modulation of important bone markers in peri-implant bone tissue.

## 1. Introduction

Diabetes mellitus (DM) is defined by the presence of hyperglycemia, and it is characterized by a heterogeneous group of metabolic alterations [1], promoting numerous complications in the quality of life and longevity. According to the International Diabetes Federation (IDF), it is probable that 425 million people present diabetes mellitus worldwide [2].

Diabetic subjects present some frequent oral manifestations, such as xerostomia, tooth caries, periodontal disease, oral candidiasis, burning mouth syndrome, altered taste, oral lichen planus, recurrent aphthous stomatitis, alterations in wound healing, increased tendency for infections and salivary-gland dysfunctions [3,4]. Some biological mechanisms justify the presence of oral manifestation in these patients, including microangiopathy in the gingiva, changes in the oral microflora, host-immune-response alteration, alterations in collagen turnover, alterations in polymorphonuclear-leukocyte function and presence of advanced-glycosylation end products [5,6,7,8,9]. Considering the elevated prevalence of DM in the population and the higher occurrence of periodontitis and successive tooth loss in these individuals [10,11,12], the use of rehabilitation with dental implants for dental replacement in diabetics is essential to effectively re-establish the occlusal functions, aesthetics and quality of life of these patients [13].

Implant therapy is a safe and predictable approach to the rehabilitation of complete and partial edentulism [14,15,16]. Nevertheless, some disorders, such as hyperglycemia related to DM, may promote harmful effects on osseous repair and compromise bone quality [17,18,19,20]. Innumerous studies have demonstrated that the diabetic condition has also a negative impact on the bone tissue around dental implants, impairing the implant stability during the healing period and promoting an increasing trend of implant complications and marginal bone loss [21,22,23,24,25].

Thus, the investigation of more predictable implant therapies in diabetic patients would be relevant to reverse the harmful influence of DM on bone healing. In this context, new bone-drilling procedures during implant placement and innovative implant macrogeometries with strategic spaces between the implant surface and the surgical bed (“healing chambers”) have been suggested as approaches able to benefit from the results related to peri-implant repair [26,27,28]. The biological mechanisms associated with the benefits promoted by these modifications of the implant macrogeometry are based on the fact that blood fills the region of the healing chambers immediately after implant insertion, contributing to the healing process [28,29,30]. Although the threads of dental implants with healing-chamber configurations do not focus on primary stability, this therapeutic strategy has been reported as an important aspect of secondary stability [28,31,32]. In fact, while data from previous investigations have suggested that the close interaction between the implant surface and bone bed is substantial for primary stability and subsequent satisfactory osseointegration [33,34], evidence has indicated that this condition may promote damage to peri-implant bone healing via the extensive bone resorption that occurs around the implant during healing [32,35,36].

In this context, information regarding the release pattern of some bone markers, such as β-catenin, Dkk1, Runx2, BMP-2, OPN and RANKL/OPG, could help to understand some molecular mechanisms related to the effect of modified implant macrodesign in the presence of DM. Briefly, β-catenin and OPN have osteogenic properties [37]. Runx2 is considered the main transcription factor of osteoblasts [38]. OPG has the key roles of binding itself to RANKL, blocking the RANK/RANKL ligation and, consequently, preventing osteoclast differentiation [39,40]. RANKL and OPG are crucial molecules for bone-homeostasis maintenance and osseous-healing control [39]. Dkk1 is considered as a potent Wnt antagonist which blocks Wnt/β-catenin signaling and as a negative regulator of osteoblast functions [41,42].

Considering these aspects, implants with optimized macrogeometry could benefit therapies with dental implants in the presence of diabetes. Thus, this study assessed, for the first time, the influence of an implant with modified macrodesign based on the presence of a healing chamber in the pattern of peri-implant repair under diabetic conditions. A better understanding of the biomechanical aspects and molecular mechanisms related to the use of modified implant macrodesign in DM could support the use of this strategy to favor the rehabilitation with dental implants in diabetic individuals.

## 2. Materials and Methods

### 2.1. Animals

The experimental practices were approved by the Animal Care and Use Committee of the university (permit number 6272060319) and followed the ARRIVE guidelines. The animal cohort comprised 10-week-old male Wistar rats (n = 30) with a mean weight of 342 ± 89 g. The acclimatization period started 15 days before the procedures, in temperature-controlled cages (24 °C) and with exposition to a 24-h light–dark cycle of equal time lengths. Water and food were accessible ad libitum (Labina, Purina1; Paulinia, SP, Brazil) in the bioterium of the university.

### 2.2. Treatment Groups

Animals were assigned to Non-DM and DM groups. Each animal received two implants, one in each tibia, at the proximal aspect of the tibial metaphysis, according to the following groups: non-diabetic animals receiving implant with conventional macrogeometry (Non-DM + Control Implant; n = 15); non-diabetic animals receiving implant with modified macrodesign with healing chambers (Non-DM + Test Implant; n = 15); animals with induced DM receiving implant with conventional macrogeometry (DM + Control Implant; n = 15); animals with induced DM receiving implant with modified macrodesign with healing chambers (DM + Test Implant; n = 15).

### 2.3. DM Induction

The fasting period to which the animals were subjected before DM induction was 14 h long. DM was induced by intraperitoneal injection of streptozotocin (STZ; Sigma-Aldrich, St. Louis, MO, USA) (60 mg/kg) dispersed in citrate buffer (Hexis, Jundiaí, SP, Brazil) (0.01 M, pH 4.5) [43] in 15 rats. An intraperitoneal injection of the same volume of 0.1 mol/L citrate buffer was given to the Non-DM rats. After 72 h, blood samples were collected from the tail of the rats and put immediately on test strips for glucose evaluation utilizing a glucose meter (Accu-Check Active^®^; Accu-Chek Active, Roche, Jacarepaguaá, RJ, Brazil). Rats that presented glucose concentrations above 300 mg/dL were considered diabetics [44].

### 2.4. Implant Placement

Briefly, under anesthesia induced using intramuscular administration of ketamine hydrochloride (10 mg/kg) and xylazine hydrochloride (Francotar, Virbac Laboratories, Roseira, SP, Brazil) (10 mg/kg), an incision of 1.0 cm in length was performed to access the tibial bone surfaces by rhombus dissection (Figure 1a). An implant bed was prepared bicortically at a rotary speed less than or equal to 1500 rpm (Figure 1b). The titanium implant used consisted of a screw-shaped, commercially available pure aluminum oxide sandblasting medium of 4.0 mm in length and 2.2 mm in diameter; this was placed in each tibia and considered inserted when the screw threads were completely fixed in the bone cortex [45] (Figure 1c). The test implant presented channels transversal to the implant threads, while the control implant did not present the channels (Implacil de Bortoli, São Paulo, SP, Brazil) (Figure 2). Then, the soft tissues were repositioned and sutured.

### 2.5. Post-Operative Period

The animals were monitored daily during the study. Thirty days after the start of the experiment, the animals were euthanized using CO_2_ inhalation. Afterwards, the dissection of the tibiae was performed to expose the implants, which were removed for a torque-force assessment. Then, the bone tissue around the implants was collected and placed in RNAlater (Ambion Inc., Austin, TX, USA) for gene-expression analyses.

### 2.6. Torque-Force Evaluation for the Removal of Implants

Following implant exposition, the connection of a torquemeter (Mark-10; BGI, New York, CA, USA) with a scale range of 0.1–10 N/cm and divisions of 0.05 N/cm was performed. It was joined, using a wrench, with the head of the implant to apply torque in the reverse direction of implant insertion until complete rupture of the bone–implant interface was shown by implant rotation. Torque-force values achieved in N/cm were considered as the torque needed for the breakdown of osseointegration [46].

### 2.7. Gene-Expression Analysis

The peri-implant bone tissue sample was kept in RNAlater at −70 °C for the evaluation of the mRNA levels of β-catenin, Dkk1, Runx2, BMP-2, OPN and RANKL/OPG. The Trizol method was used to isolate total RNA from the bone biopsies (Gibco BRL; Life Technologies, Rockville, MD, USA), according to the manufacturer’s instruction and as previously described [47]. At first, diethylpyrocarbonate-treated water was used to resuspend RNA that was stored at −70 °C. A micro-volume spectrophotometer (Nanodrop 1000; Nanodrop Technologies LLC, Wilmington, NC, USA) was used to define the RNA concentration.

DNase (Turbo DNA-frees; Ambion Inc., Austin, TX, USA) was used to treat total RNA, which was used for complementary DNA (cDNA) synthesis (First-Strand cDNA synthesis kit—Roche Diagnostic Co., Indianapolis, IN, USA), according to the manufacturer’s instructions. Software of prob design was used to design the primers (Light-Cycler Roche probe design software; Diagnostics GmbH, Mannheim, Germany) (Table 1). Quantitative real-time polymerase chain reaction (qPCR) was performed in real-time PCR equipment (LightCyclers Software 4; Roche Diagnostics GmbH, Mannheim, Germany) employing a Syber Green kit (FastStart DNA Masterplus Syber Green; Roche Diagnostic Co., Indianapolis, IN, USA). The results were expressed as relative amounts of the target gene using GAPDH as the inner reference gene using the relative quantification tool, following the manufacturer’s recommendations.

### 2.8. Data Analyses

The data were examined for normality using the Kolmogorov–Smirnov test. Since the torque-force data achieved normality, parametric methods were used for the comparisons (SAS software—Program Release 9.3; Cary, NC, USA). Then, a Split-Plot Analysis of Variance was used for the comparison of the biomechanical-retention analysis results of the titanium implants. Gene-expression data did not reach normality; non-parametric methods were applied for the analyses, and the Wilcoxon signed-rank test and Mann–Whitney U test were used for the comparison of the differences in the relative gene expression. A significance level of 5% was set for all analyses.

## 3. Results

### 3.1. Torque-Force Evaluation

The biomechanical analyses showed that DM promoted a negative impact on implant torque levels independently of implant macrodesign, showing reduced counter-torque values in comparison with Non-DM rats (*p* < 0.05). In addition, the modified implant macrogeometry promoted promising outcomes for the biomechanical retention of the titanium implants, achieving higher torque levels in Non-DM than in DM rats (*p* < 0.05). Figure 3 illustrates the counter-torque-force values of each group.

### 3.2. Gene-Expression Levels

The gene-expression evaluation revealed that both OPN and β-catenin concentrations were higher in the bone tissue around the implants of Non-DM animals than in DM rats, independently of implant thread design (*p* < 0.05). It was also observed that the new implant macrodesign promoted a positive impact on the mRNA levels of OPN in the peri-implant bone tissue, upregulating its concentrations in Non-DM and DM rats (*p* < 0.05). The mRNA levels of Dkk1, Runx2, BMP-2, RANKL, OPG and RANKL/OPG did not present significant differences among groups (*p* > 0.05). Figure 4 illustrates the mRNA quantification of all genes evaluated in each group.

## 4. Discussion

Tooth rehabilitation using implant therapy in the presence of diabetes is an essential approach able to offer satisfactory nutritional intake and efficient chewing, which may significantly favor the life quality of these individuals [16]. Complications attributed to uncontrolled glycemic levels are observed in several diabetic individuals, even though the efforts to regulate hyperglycemia impair the osteoimmunoregulatory response and promote a harmful influence on bone healing [18,21,23,25,47,48,49,50,51,52]. Considering that modifications of the implant macrogeometry based on the presence of healing chambers may benefit the healing process around implants [28,29], this study aimed to establish the impact of an implant with modified macrogeometry on bone repair and on the release of bone-related markers in this tissue around implants in animals with induced DM. In general, the data from this study showed that the new implant design optimized the biomechanical behavior in diabetic circumstances, improving the molecular performance of the bone tissue around implants.

Of interest, and in agreement with the results achieved in this experimental study, previous data have already verified that the existence of healing chambers may improve osseointegration, significantly modifying the biological peri-implant healing pattern under systemically healthy conditions [25,27,28,52,53]. The healing-chamber concept has been introduced to the dental-implant design of threads in the last decade [33,53], and recent studies have confirmed the biological impact of this approach [28]. In fact, previous investigations under systemically healthy conditions have supported that alterations in the macrodesign of dental implants incorporating healing chambers in the implant body promote promising outcomes in terms of peri-implant bone healing, since the spaces between the healing-chamber device and the residual socket walls are preserved and rapidly occupied by a blood clot during implant placement, which improves bone-healing progression by forming osteogenic tissue, causing woven bone ingrowth [28,29,30,31,32,52].

Although data from a recent experimental investigation assessing the histometric and biomechanical impacts of implant healing chambers have shown no benefits of this approach to bone-to-implant contacts, the authors have reported promising results in terms of bone density when comparing healing chambers with implants with conventional configurations [54]. In agreement with these findings, other experimental studies have shown that the implant with a chamber design may promote better wound healing and stimulate early bone formation inside the threads of implants [55].

Even though many investigations have shown positive performance with the use of implant healing-chamber configurations [31,32,33,52,54], no data concerning the performance of modified implant macrodesign in the presence of systemic conditions related to harmful bone healing, such as hyperglycemia, are available. Thus, to our knowledge, this research study is the first to analyze the role of a modified implant macrogeometry on bone healing around implants under diabetic conditions.

Biomechanical assessments are usually recommended as measurable indicators of the force required by a torque to collapse the bone–implant interface near the implant surfaces. According to our data, the biomechanical analyses of the implants showed that, although DM animals presented impaired torque-force levels compared with Non-DM animals (*p* < 0.05), augmented resistance to removal torque was achieved by the implants with modified macrogeometry in comparison with conventional-configuration implants (*p* < 0.05). These encouraging outcomes seen in this experiment suggest the benefits of using a modified macrodesign implant as a therapeutic strategy to improve peri-implant bone healing, even under challenging circumstances for bone repair, such as the presence of diabetes mellitus. Although no information concerning the effect of implant healing chambers under hyperglycemic conditions is available, previous data have detected augmented implant biomechanical fixation in implants with healing chambers when compared with the traditional thread design [56], in line with the current findings.

Interestingly, the encouraging outcomes verified in the current study concerning the biomechanical retention of titanium implants achieved by modified implant macrogeometry under hyperglycemic circumstances may be clarified, in part, by the modulation of host osteoimmune response during peri-implant healing processes. Our results reveal that the presence of DM negatively modulated the gene expression of OPN and β-catenin in bone tissue, diminishing the concentrations of these bone-related markers in osseous biopsies as compared with Non-DM controls (*p* < 0.05). In line with our results, earlier investigations have confirmed the reduced levels of these molecules in the samples of bone tissue under hyperglycemic conditions, indicating that impaired bone healing in the presence of DM could be associated, at least in part, with the modulation of these biomarkers in bone tissue [23,57,58,59,60].

Interestingly, in this research study, the gene-expression assay of the bone tissue surrounding the implants demonstrated that the use of a modified implant macrodesign in DM animals promoted the upregulation of the OPN levels with a significative augment when compared with samples from the DM animals that had received conventional implants (*p* < 0.05), as also observed in Non-DM animals (*p* < 0.05). It is well known that OPN is a crucial molecule related to bone metabolism, acting on the recruitment and differentiation of undifferentiated mesenchymal stem cells, stimulating angiogenesis and promoting tissue mineralization [61,62,63,64]. Earlier investigations have even reported the growth of non-collagenous molecules of the organic matrix, such as OPN, at the bone–implant limits [65,66,67], encouraging primary-mesenchymal-stem-cell migration around the peri-implant environment in a dose-dependent way and promoting the differentiation of these cells toward an osteogenic lineage [62,68,69,70].

In this context, it is essential to highlight that the presence of a healing chamber in implants with modified macrodesign may favor the concentration of osteocytes in the region, accomplishing higher amounts of newly formed bone tissue inside the thread zones [54]. Furthermore, the healing-chamber configuration can accelerate the development stage of bone peri-implant repair, as recently supported by the existence of osteons enclosed by lamellar components with centric osteocytes promoting an anticipated organization of bone tissue in implants with a healing chamber [55]. Importantly, it has been described that the biological advantages provided by healing chambers are also related to the early mineralization process detected in subsequent bone-healing phases [55].

Increased OPN has been frequently seen during matrix maturation in bone tissue, indicating the binding of basic components to the extracellular bone matrix [67,71], although all the mentioned data may partially support our molecular results showing that the use of an implant with modified macrogeometry can upregulate the mRNA expression of OPN in the bone biopsies from DM rats and, subsequently, contain the harmful influence of diabetes on peri-implant bone tissue. Nevertheless, there is a paucity of data in the literature evaluating the role of implants with healing chambers on the local modulation of bone-related markers around dental implants, especially under diabetic challenge. Additional studies would be important to support the data presented in the current investigation and to better elucidate the molecular mechanisms associated with the use of modified implant macrodesign in this at-risk condition for peri-implant repair. It is essential to highlight that our data reveal molecular outcomes related to later events of osseointegration, and further studies evaluating short-term periods of peri-implant bone repair would clarify the influence of the healing-chamber configuration on previous stages of repair around implants.

Noteworthy, a recent experimental study on diabetic rats has demonstrated that, although treatment with hypoglycemic medications may offer some molecular benefits in peri-implant repair, it was not enough to histometrically reverse the damaging influence of hyperglycemia on the osseointegration process [72]. Other alternative treatments to surpass the harmful effect of hyperglycemia on bone repair around implants, such as the use of the parathyroid hormone, mesenchymal-stem-cell therapy, hyperbaric-oxygen management, doxycycline administration and implant-surface modification, have been evaluated [37,51,73,74,75,76,77]. Nonetheless, some of these therapeutic strategies have been associated with adverse effects, and most of them have offered unpredictable results on peri-implant osseointegration, without overcoming the harmful influence of diabetes on bone repair around implants.

Individuals presenting poorly controlled diabetes are more susceptible to damaged osseointegration, are at higher risk of implant failure and have a greater chance of peri-implant lesions [22,25]. Therefore, taking into account the hopeful outcomes obtained in the current investigation with a modified implant macrogeometry in diabetic rats, its use could be suggested as an alternative to containing the detrimental effect of DM on bone repair around implants, besides reducing the possibility of peri-implant clinical complications in diabetic patients. The use of implants with a healing-chamber configuration seems to be relevant, especially under systemic and local at-risk conditions, to peri-implant repair. However, additional evidence should be established to validate the results of this study under diabetic conditions, considering that this is the first one, to the authors’ understanding, to assess the role of a modified implant macrodesign with healing chambers to contain the negative result of hyperglycemia on peri-implant bone repair.

Some limitations of this study can be highlighted. A histomorphometric analysis could have reinforced the results obtained. On the other hand, the counter-torque analysis performed is a reliable tool for biomechanical assessments, capable of measuring the force required by a torque to collapse the bone–implant interface surrounding the implant surfaces, which could be comparable to the evaluation of the bone-to-implant contact obtained with a histological analysis. Future studies encompassing micro-tomography or even micro-magnetic resonance imaging, which features a 3D analysis in addition to a histomorphometric analysis, would be important to improve the findings of implant or biomaterial studies [78]. Additionally, the quantification of proteins related to the studied genes would help to comprehend the molecular process related to the effect of the presence of a healing chamber on implants with modified macrodesign under diabetic conditions. Nevertheless, all analyses conducted in this study provided essential information concerning the objectives proposed by the investigation.

Within the limits of this study, it can be concluded that the modified macrogeometry of implants may benefit peri-implant repair under diabetic conditions, promoting higher biomechanical retention and favoring the release of important bone markers in the bone tissue around implants.

Abbreviations used in this paper: qPCR, quantitative real-time polymerase chain reaction; bp, base pairs; Runx2, runt-related transcription factor 2; BMP-2, bone morphogenetic protein 2; OPN, osteopontin; Dkk1, Dickkopf 1; RANKL, receptor activator of the nuclear factor kappa B ligand; OPG, osteoprotegerin; GAPDH, glyceraldehyde-3-phosphate dehydrogenase.

## Figures and Tables

**Figure 1 materials-15-02317-f001:**
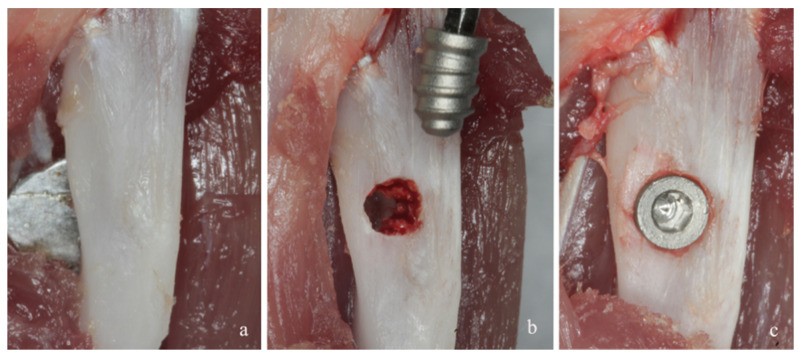
(**a**) Surgical exposition of the bone surface of the tibiae by rhombus dissection. (**b**) Bicortically prepared implant bed and placement of titanium implant into the tibia. (**c**) Implant was completely inserted when all screw threads had fixed into the bone cortex.

**Figure 2 materials-15-02317-f002:**
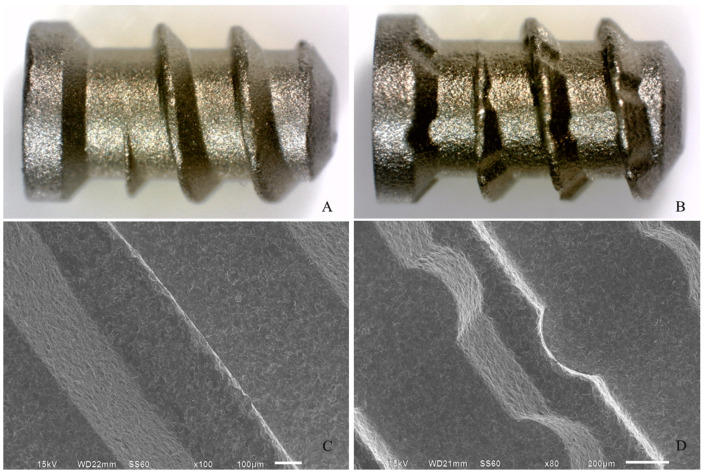
(**A**,**B**) Stereo microscopy of the implants (3×) for control and test groups, respectively. (**C**,**D**) Scanning electron microscopy of the implants (80–100×) for control and test groups, respectively.

**Figure 3 materials-15-02317-f003:**
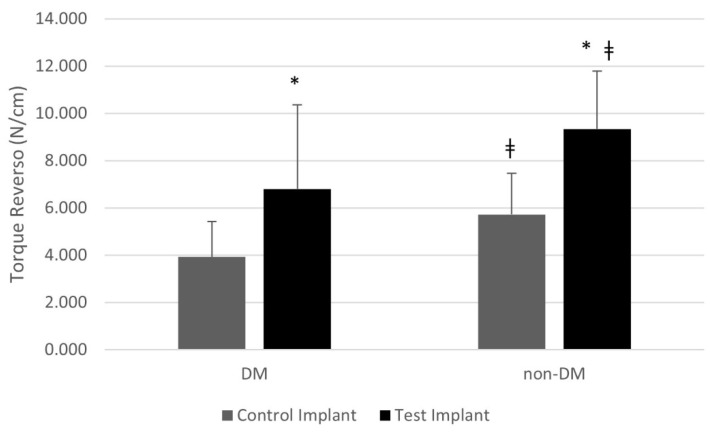
* indicates intra-group statistical differences (Split-Plot Analysis of Variance; *p* < 0.05). ǂ indicates inter-group statistical differences (Split-Plot Analysis of Variance; *p* < 0.05).

**Figure 4 materials-15-02317-f004:**
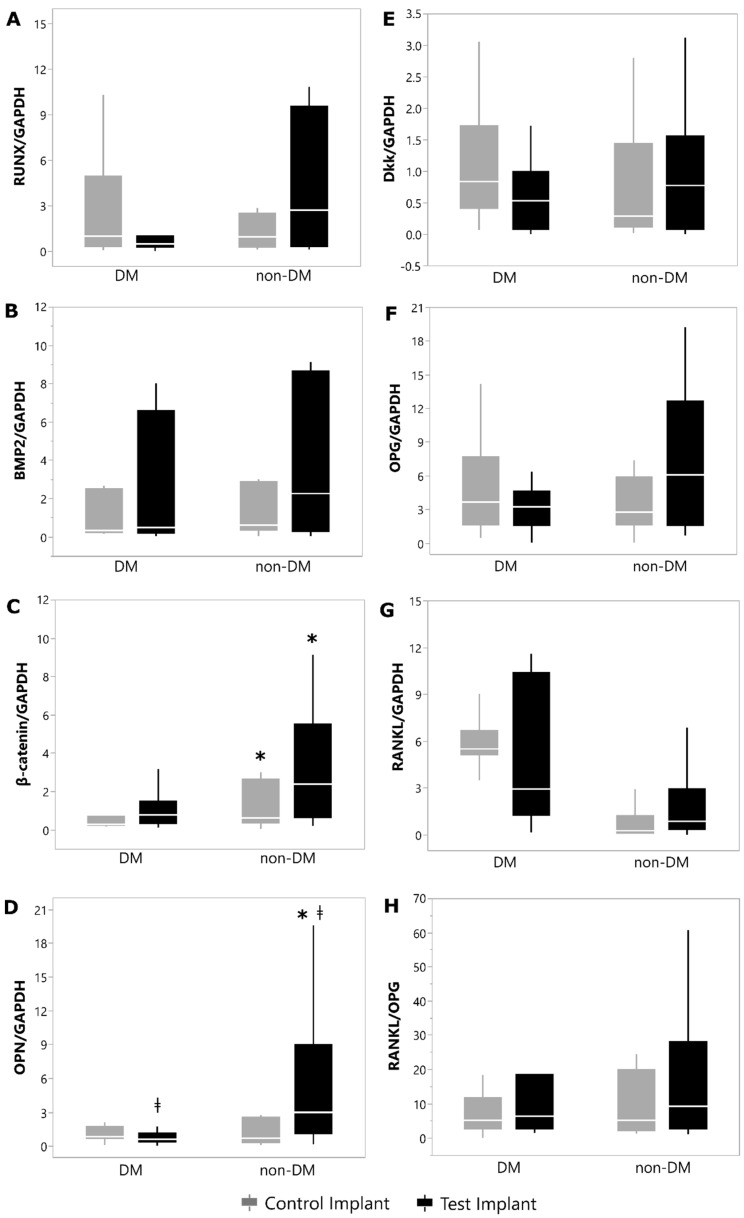
ǂ indicates intra-group statistical differences (Wilcoxon Signed-rank Test; *p* < 0.05). * indicates inter-group statistical differences (Mann–Whitney U Test; *p* < 0.05). Relative levels of mRNA for: (**A**)—RUNX/GAPDH; (**B**)—BMP-2/GAPDH; (**C**)—β-CATENIN/GAPDH; (**D**)—OPN/GAPDH; (**E**)—OPG/GAPDH; (**F**)—DKK/GAPDH; (**G**)—RANKL/GAPDH; (**H**)—RANKL/OPG.

**Table 1 materials-15-02317-t001:** Primer sequences, amplification profiles and the estimated length values of the qPCR product for each gene.

Gene	Sequence (5′–3′)	Length of qPCR Product (bp)	Amplification Profile (Temperature, Time)
Runx2	GCCACTTACCACAGAGC	157	95 °C, 10 s/56 °C, 8 s/72 °C, 7 s
BMP-2	GTCCCTACTGATGATGAGTTTCTC	170	95 °C, 10 s/56 °C, 8 s/72 °C, 8 s
OPN	CCGGATGCAATCGATAGTG	164	95 °C, 10 s/56 °C, 7 s/72 °C, 8 s
β-catenin	ACTCTGAGAAACTTGTCCG	172	95 °C, 10 s/56 °C, 8 s/72 °C, 8 s
Dkk1	CGGGAATTACTGCAAAAACG	83	95 °C, 9 s/59 °C, 9 s/72 °C, 9 s
RANKL	AGCGCTTCTCAGGAGTT	156	95 °C, 5 s/55 °C, 4 s/72 °C, 6 s
OPG	GCAGAGAAGCACCTAGC	168	95 °C, 10 s/56 °C, 8 s/72 °C, 7 s
GAPDH	TGAGTATGTCGTGGAGTCTACTG	159	95 °C, 10 s/56 °C, 8 s/72 °C, 7 s

qPCR, quantitative real-time polymerase chain reaction; bp, base pair; Runx2, runt-related transcription factor 2; BMP-2, bone morphogenetic protein 2; OPN, osteopontin; Dkk1, Dickkopf 1; RANKL, receptor activator of the nuclear factor kappa B ligand; OPG, osteoprotegerin; GAPDH, glyceraldehyde-3-phosphate dehydrogenase.

## Data Availability

The data presented in this study are available on request from the corresponding author.

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
