# Peer review of "Peri-Implant Repair Using a Modified Implant Macrogeometry in Diabetic Rats: Biomechanical and Molecular Analyses of Bone-Related Markers"

_materials, 2022, doi:10.3390/ma15062317_

Round 1
Reviewer 1 Report
The study shows better mechanical strength and higher OPN expression in DM rats implanted with healing chambers compared to standard implants. The data presented are rather limited, and not supported by imaging or histomorphometric results. The absence of these data should be justified or listed as the major limitation of this study. The term osteoimmune response is coined but there are no data to support that. I also do not agree with the statistical analysis used. Split plot analysis of variance indicates time-dependent changes, which was not the case for this study. I think factorial ANOVA is a better approach (DM status X type of implants).
Other minor comments are as the following: - pay attention to the citation format, the numbers are not in brackets and they are very confusing.
- line 73: "30 10-week old male Wistar rats..." may be confusing at first glance. suggest changing to "10-week-old male Wistar rats (n=30)"
- line 75: temperature-controlled - which temperature?
- line 94: provide a citation for the hyperglycaemic cut-off 300 mg/dL.
- line 115: "the rats are accompanied..." do the authors mean "monitored"
section 2.6: it will be beneficial to the readers to have a picture to depict the procedures - line 163: Wilcoxin sign-ranked test and Mann Whitney U-test
- Figure 4: I disagree with the presentation of the graphs. Skewed data should not be presented as mean and SD. Please use box plot or scatterplot instead. Typo for title 'beta-catenin"
- Discussion: The authors should include limitations of the study.
Author Response
Response to referees
Reviewer 1
- The study shows better mechanical strength and higher OPN expression in DM rats implanted with healing chambers compared to standard implants. The data presented are rather limited, and not supported by imaging or histomorphometric results. The absence of these data should be justified or listed as the major limitation of this study. The term osteoimmune response is coined but there are no data to support that. I also do not agree with the statistical analysis used. Split plot analysis of variance indicates time-dependent changes, which was not the case for this study. I think factorial ANOVA is a better approach (DM status X type of implants).
We thank the reviewer comments and suggestions. The authors listed the absence of data regarding histomorphometric results as a limitation of the study in the discussion section. We have chosen to conduct gene expression and counter-torque analysis due to the sample processing. When the counter-torque is done, the bone tissue is collected for gene expression analysis without compromising the sample. When the histological procedure is carried out, the process precludes the use of the same sample for gene expression analysis. The counter-torque analysis performed is a reliable toll for biomechanical assessments capable of measure the force that a torque requires to collapse the bone-implant interface surrounding implant surfaces, which could be comparable to the bone-to-implant contact evaluation. Besides, considering that this study is an in vivo methodology, it would be unethical to unnecessarily increase the sample size.
The authors also replaced the term osteoimmune response for “bone markers” (Peri-implant bone repair may profit from the use of implants with modified macrogeometry in the presence of diabetes mellitus, with higher biomechanical retention, and positive modulation of important bone markers in peri-implant bone tissue.).
The split plot analysis of variance can be used to determine if two or more repeated measures from 2 or more groups are significantly different from each other on your variable of interest. In our study, although we don’t have different time-period, two different treatments were administered in each animal, such as a split mouth study. Therefore, the used statistical test is in accordance with the study design.
- Other minor comments are as the following: - pay attention to the citation format, the numbers are not in brackets, and they are very confusing.
The references were corrected, and the text was revised regarding grammar and spelling.
- line 73: "30 10-week old male Wistar rats..." may be confusing at first glance. suggest changing to "10-week-old male Wistar rats (n=30)"
The modification was done in the phrase.
- line 75: temperature-controlled - which temperature?
The room temperature is 24ºC. The information was added to the text.
- line 94: provide a citation for the hyperglycemic cut-off 300 mg/dL.
A citation was provided:
Kador PF, O'Meara JD, Blessing K, Marx DB, Reinhardt RA. Efficacy of structurally diverse aldose reductase inhibitors on experimental periodontitis in rats. J Periodontol. 2011 Jun;82(6):926-33. doi: 10.1902/jop.2010.100442. Epub 2010 Dec 28. PMID: 21189083.
- line 115: "the rats are accompanied..." do the authors mean "monitored"
The authors thank the reviewer observation, and the term was modified as suggested.
- section 2.6: it will be beneficial to the readers to have a picture to depict the procedures
Surgical procedures are shown at figure 1.
- line 163: Wilcoxon sign-ranked test and Mann Whitney U-test.
The tests were described as indicated.
- Figure 4: I disagree with the presentation of the graphs. Skewed data should not be presented as mean and SD. Please use box plot or scatterplot instead. Typo for title 'beta-catenin"
The authors modified the graph stile as recommended.
- Discussion: The authors should include limitations of the study.
The authors agree and thank the reviewer. We added some limitations of the study as above:
Some limitations of this study can be highlighted. The presence of histomorpho-metric analysis could reinforce the results obtained. On the other hand, the coun-ter-torque analysis performed is a reliable tool for biomechanical assessments capable of measuring the force required by a torque to collapse the bone-implant interface sur-rounding implant surfaces, which could be comparable to the bone-to-implant contact evaluation obtained in a histological analysis. Future studies encompassing mi-cro-tomography or even micro- Magnetic Resonance Imaging, which features 3D anal-ysis additional to histomorphometric analysis are important to improve the findings of implant or biomaterial studies [80]. Additionally, the quantification of proteins related to the studied genes would help comprehend the molecular process related to the effect of the presence of a healing chamber in implants with modified macrodesign in diabetic conditions. Nevertheless, all analyses conducted in this study provided essential in-formation concerning the objectives proposed by the investigation.

Reviewer 2 Report
In this study, the authors have investigated the gene expression of bone-related markers in Non-DM vs DM in rats that had implants (Conventional microgeometry as a control vs Modified geometry as a test group). The authors also demonstrated that the expression level of OPN and β-catenin were higher in Non-DM vs DM group. In addition, the results showed there were no significant differences in the expression level of the other bone-related markers tested. On the basis of these observations, the authors concluded that the modified macrogeometry of implants may benefit the peri-implant repair in diabetic rats.
The overall focus of the manuscript is good and would be of interest. However, the following points need to be addressed:
- The authors should add a brief description of the functions of the bone-related markers tested in this paper in the introduction.
- For how long have the rats been diabetic before implants placement?
- The authors should elaborate more on the difference between conventional vs modified geometry in the methods
- Figure 4, labeling the panels in to A, B, C ….itc would be easier to follow
- Figure 4, correct the spelling of β-catenin.
- The discussion should end with the authors conclusions based on their findings
Author Response
Response to referees
Reviewer 2
In this study, the authors have investigated the gene expression of bone-related markers in non-DM vs DM in rats that had implants (Conventional microgeometry as a control vs Modified geometry as a test group). The authors also demonstrated that the expression level of OPN and β-catenin were higher in non-DM vs DM group. In addition, the results showed there were no significant differences in the expression level of the other bone-related markers tested. On the basis of these observations, the authors concluded that the modified macrogeometry of implants may benefit the peri-implant repair in diabetic rats.
The overall focus of the manuscript is good and would be of interest. However, the following points need to be addressed:
- The authors should add a brief description of the functions of the bone-related markers tested in this paper in the introduction.
The authors thank the reviewer comments and added to the introduction section the description of the functions of the bone-related markers tested:
In this context, information regarding the release pattern of some bone markers, such as β-catenin, Dkk1, Runx2, BMP-2, OPN, and RANKL/OPG could help to under-stand some molecular mechanisms related to the effect of modified implant macrodesign in the presence of DM. Briefly, β-catenin and OPN have osteogenic prop-erties [38]. Runx2 is considered the main transcription factor of osteoblasts [39]. OPG has a key role biding itself to RANKL, blocking the RANK/RANKL ligation and con-sequently, preventing osteoclast differentiation [40, 41]. RANKL and OPG are crucial molecules for bone homeostasis maintenance and osseous healing control [40]. Dkk1 is considered as a potent Wnt antagonist which blocks Wnt/β-catenin signaling and as a negative regulator of osteoblast function [42, 43].
- For how long have the rats been diabetic before implants placement?
The implant insertion surgery occurred at the third day after diabetes induction. Although this model of diabetes is considered an acute model, it is generally used to investigate cellular and tissue alterations. The streptozotocin induction model has been used to study changes like:
- wound healing (Al-Bayaty & Abdulla 2012, Toung et al 2012)
- Oxidative stress activity (Aziz et al 2012, Faria et al 2012)
- Liver activity and lipid profile (Krishnakumari et al 2011, Abdel et al 2012)
- Nephropathy (Banki et al 2012, Friederich et al 2012, Peng et al 2012, Zhang et al et al 2012, Sun et al 2012, Wang et al 2012, Korropati et al 2012)
- Cardiovascular disease (Salum et al 2012, Li et al 2012, Jankyova et al 2012, Murça et al 2012, Xu et al 2012, Fallahi et al 2012, Delucchi et al 2012, Pari et al 2012, Quintela et al 2012)
- Collagen degradation (Eliezer et al 2012)
- Retinopathy (Lu et al 2012, Guinta et al 2012, Lio et al 2012, Pan et al 2012)
- Testicular dysfunction (Tsounapi et al 2012, fernandes et al 2011, Xu et al 2010, Ricci et al 2009)
- Neuropathy (Huang et al 2012, Kim et al 2012, Yamazaki et al 2012, Vargas et al 2012).
- The authors should elaborate more on the difference between conventional vs modified geometry in the methods.
The authors thank the reviewer suggestion, and the description was rewritten:
A screw-shaped, commercially available pure aluminum oxide sandblasting titanium implant, 4.0 mm in length and 2.2 mm in diameter was placed in each tibia until the screw threads had been completely embedded in the bone cortex32 (Figure 1c). The test implant presented channels transversal to the implant threads, while the control implant did not present the channels (Implacil de Bortoli, São Paulo, SP, Brazil) (Figure 2). Then, the soft tissues were replaced and sutured.
- Figure 4, labeling the panels in to A, B, C ….it would be easier to follow
The figure was modified to clarify the understanding.
- Figure 4, correct the spelling of β-catenin.
The spelling was corrected.
- The discussion should end with the authors conclusions based on their findings
Within the limits of this study, it can be concluded that the modified macrogeometry of implants may benefit the peri-implant repair in diabetic conditions, promoting higher biomechanical retention and favoring the release of important bone markers in the bone tissue around implants

Reviewer 3 Report
The article entitled “Peri-implant repair using a modified macrogeometry of implant in diabetic rats: Biomechanical and gene expression analysis of bone-related markers” aimed to evaluate the influence of an implant with modified macrodesign based on the presence of healing chamber in the patter of peri-implant repair under diabetic conditions. The paper is in line with journal’s aim, moreover, Authors have well revised several issues; however, I ask authors to add some key concepts.
- In the title of the article, authors should specify the type of study
- The introduction section is too short, the author must add and discuss previous findings regarding the most frequent oral manifestations in subjects with type 1 and type 2 diabetes mellitus (please see and discuss doi: 10.3390 / jcm9072196)
- Table 1 is confusing, please, rewrite it
- Figures 3 and 4 are not very sharp, they must be replaced with other more defined tables
- The results are too long, they need to be shortened
- The discussion must be implemented, both in the part of the bibliography and in the analysis of the results obtained, why not discuss other tools for bone and implant morphology analysis? (Not only in vitro, please see and discuss doi: 10.1002 / term.2494)
- The limits of the study should be included in the paper
- Conclusions cannot be reduced to a sentence: you must improve them highlighting the limits and the future insights pointed out from this article.
- The formatting of the references is not correct, please check the journal instructions for authors
- Several moderate typos are present in the text, please, amend
According to this Reviewer’s consideration, novelty and quality of the paper, publication of the present manuscript is recommended after minor revision.
Author Response
Response to referees
Reviewer 3
The article entitled “Peri-implant repair using a modified macrogeometry of implant in diabetic rats: Biomechanical and gene expression analysis of bone-related markers” aimed to evaluate the influence of an implant with modified macrodesign based on the presence of healing chamber in the patter of peri-implant repair under diabetic conditions. The paper is in line with journal’s aim, moreover, Authors have well revised several issues; however, I ask authors to add some key concepts.
- In the title of the article, authors should specify the type of study.
The authors thank and the title was modified following the reviewer suggestion:
“Peri-implant repair using a modified macrogeometry of implant in diabetic rats: Biomechanical and gene expression analysis of bone-related markers”. An animal study.
- The introduction section is too short, the author must add and discuss previous findings regarding the most frequent oral manifestations in subjects with type 1 and type 2 diabetes mellitus (please see and discuss doi: 10.3390 / jcm9072196)
The authors thank the reviewer suggestion. Some points were added to the discussion section as well as the citation recommended.
Diabetic subjects present some frequent oral manifestations, such as xerostomia, tooth caries, periodontal disease, oral candidiasis, burning mouth syndrome, altered taste, oral lichen planus, recurrent aphthous stomatitis, alteration in wound healing, increased tendency for infections, and salivary gland dysfunction [3,4]. Some biological mechanisms justify the presence of oral manifestation in these patients: microangiop-athy in the gingiva; changes in the oral microflora; host immune response alteration; alteration of collagen turnover; alteration of polymorphonuclear leukocytes function; the presence of advanced glycosylation end products [5-9].
- Table 1 is confusing, please, rewrite it
The table was rewritten.
- Figures 3 and 4 are not very sharp, they must be replaced with other more defined tables.
Both figures were reformulated.
- The results are too long, they need to be shortened.
The results were rewritten as requested.
- The discussion must be implemented, both in the part of the bibliography and in the analysis of the results obtained, why not discuss other tools for bone and implant morphology analysis? (Not only in vitro, please see and discuss doi: 10.1002 / term.2494).
We thank the valuable suggestions. Some points were added to the discussion section as well as the citation recommended.
Some limitations of this study can be highlighted. The presence of histomorphometric analysis could reinforce the results obtained. On the other hand, the countertorque analysis performed is a reliable tool for biomechanical assessments capable of measuring the force required by a torque to collapse the bone-implant interface sur-rounding implant surfaces, which could be comparable to the bone-to-implant contact evaluation obtained in a histological analysis. Future studies encompassing microtomography or even micro- Magnetic Resonance Imaging, which features 3D analysis additional to histomorphometric analysis are important to improve the findings of implant or biomaterial studies [80]. Additionally, the quantification of proteins related to the studied genes would help comprehend the molecular process related to the effect of the presence of a healing chamber in implants with modified macrodesign in diabetic conditions. Nevertheless, all analyses conducted in this study provided essential in-formation concerning the objectives proposed by the investigation.
- The limits of the study should be included in the paper.
The authors agree and thank the reviewer. We added some limitations of the study at the final part of discussion section, discussing other methods for bone-implant analysis as suggested before.
- Conclusions cannot be reduced to a sentence: you must improve them highlighting the limits and the future insights pointed out from this article.
Conclusions were rewritten as requested.
- The formatting of the references is not correct, please check the journal instructions for authors
Several moderate typos are present in the text, please, amend.
The references were corrected, and the text was revised regarding grammar and spelling.
- According to this Reviewer’s consideration, novelty and quality of the paper, publication of the present manuscript is recommended after minor revision.
The authors thank the referee for the interesting observations and suggestions.

Round 2
Reviewer 1 Report
Title: It is mentioned the study was conducted in diabetic rats. So the phrase "animal study" is redundant.
I still think that split-plot ANOVA is not the best choice for the analysis of biomechanical strength because the same rats are not measured twice. A factorial ANOVA looking at the effects of DM and implants will be more appropriate.
Figure 3- what is Nao DM?
For the gene expression results, I saw RANKL/OPG ratio but the OPG expression level was not presented.
Author Response
Response to referees
Reviewer 1
Title: It is mentioned the study was conducted in diabetic rats. So the phrase "animal study" is redundant.
We agree with the reviewer’s opinion. We included the term “animal study” following a request of one of the referees. Now, we will remove the term as suggested by this reviewer.
I still think that split-plot ANOVA is not the best choice for the analysis of biomechanical strength because the same rats are not measured twice. A factorial ANOVA looking at the effects of DM and implants will be more appropriate.
We appreciate the reviewer concern. To better understand the point, we have consulted a statistical expert about this issue, who pointed that our approach is in accordance with her understanding. We have used the same test applied before by our research group, in a study with the same design, in which the statistical analysis was performed by a specialist in statistics*. Split-plot design is used when pieces or wholes, which receive the levels of one of the factors, are strongholds of units in subplaces or subunits, to which the levels are an additional factor are applied. Each plot works as a “block” for the treatments of the subplots.
* Corrêa MG, Gomes Campos ML, Marques MR, Bovi Ambrosano GM, Casati MZ, Nociti FH Jr, Sallum EA. Outcome of enamel matrix derivative treatment in the presence of chronic stress: histometric study in rats. J Periodontol. 2014 Jul;85(7):e259-67.
Figure 3- what is Nao DM?
The figure was corrected (non-DM).
For the gene expression results, I saw RANKL/OPG ratio but the OPG expression level was not presented.
OPG expression was added to the results section and a graph was added to Figure 4.
